# Flexible Pressure Sensors Based on Microcrack Structure and Composite Conductive Mechanism for Medical Robotic Applications

**DOI:** 10.3390/mi14061110

**Published:** 2023-05-25

**Authors:** Qiang Zou, Yuheng Xie, Yunjiang Yin, Baoguo Liu, Yi Yu

**Affiliations:** 1School of Microelectronics, Tianjin University, Tianjin 300072, China; 2Tianjin International Joint Research Center for Internet of Things, Tianjin 300072, China; 3Tianjin Key Laboratory of Imaging and Sensing Microelectronic Technology, Tianjin 300072, China

**Keywords:** flexible electronics, pressure sensor, machine touch, medical robot, force feedback

## Abstract

With the advancement of intelligent medical robot technology, machine touch utilizing flexible sensors has emerged as a prominent research area. In this study, a flexible resistive pressure sensor was designed incorporating a microcrack structure with air pores and a composite conductive mechanism of silver/carbon. The aim was to achieve enhanced stability and sensitivity with the inclusion of macro through-holes (1–3 mm) to expand the sensitive range. This technology solution was specifically applied to the machine touch system of the B-ultrasound robot. Through meticulous experimentation, it was determined that the optimal approach involved uniformly blending ecoflex and nano carbon powder at a mass ratio of 5:1, and subsequently combining the mixture with an ethanol solution of silver nanowires (AgNWs) at a mass ratio of 6:1. This combination of components resulted in the fabrication of a pressure sensor with optimal performance. Under the pressure testing condition of 5 kPa, a comparison of the resistance change rate was conducted among samples using the optimal formulation from the three processes. It was evident that the sample of ecoflex-C-AgNWs/ethanol solution exhibited the highest sensitivity. Its sensitivity was increased by 19.5% compared to the sample (ecoflex-C) and by 11.3% compared to the sample (ecoflex-C-ethanol). The sample (ecoflex-C-AgNWs/ethanol solution), which only incorporated internal air pore microcracks without through-holes, exhibited sensitive response to pressures below 5 N. However, with the addition of through-holes, the measurement range of its sensitive response increased to 20 N, representing a 400% increase in the measurement range.

## 1. Introduction

The human body heavily relies on tactile perception to comprehend the surrounding environment, encompassing stimuli such as pressure [1,2], strain [3,4], torsion [5,6], shear force [7,8], and touch [9,10]. Among these, pressure plays a vital role in tactile perception, as mechano-receptors on the skin detect stress and provide insights into the well-being of different body parts [11,12,13,14,15]. With the emergence of wearable devices and the Internet of Things, flexible pressure sensors have gained significant traction in various fields, including electronic skin [16,17], human–machine interfaces [18,19], prosthetics [20,21], soft robots [22,23], and medical equipment [24,25]. Extensive research has been conducted to explore the potential of pressure sensors in converting external pressure into electrical signals. These sensors hold promise in detecting pressure changes and converting them into measurable electrical signals for diverse applications.

Flexible pressure sensors play a crucial role in converting external pressure stimuli into electrical signals. However, existing sensors encounter limitations in balancing high sensitivity and a wide measurement range. These sensors can be categorized into three conduction mechanisms: resistive [26,27,28,29,30], capacitive [31,32,33,34,35], and piezoelectric [36,37,38]. When aiming to achieve the lowest detection limit and resolution, striking a balance between sensitivity and range becomes challenging.

To address this challenge, this paper proposes a novel approach by combining porous microcracks and macro through-hole structures. This innovative design aims to explore new methods for achieving both high sensitivity and an extended measurement range. By incorporating these structures, the flexible pressure sensor can better cater to the tactile sensing requirements of ultrasound robots (as shown in Figure 1). This enables the utilization of a wide range and sensitive flexible pressure sensor to replicate the tactile function of a human hand in robotic applications. Furthermore, with the future integration of machine vision to replace the visual judgment of doctors, it can lead to the realization of a true B-ultrasound AI-robotic medical application.

The primary focus of this study lies in the application scenario of B-ultrasound intelligent robots. To overcome the limitations, a flexible resistive pressure sensor based on a polymer composite structure is proposed. This sensor incorporates a microcrack structure and composite conductive mechanism to enhance sensitivity and reliability. It is crucial to carefully consider the chemical composition, physical structure, and graphical distribution at the contact area with the probe of the flexible pressure sensor, ensuring that it does not interfere with the B-mode ultrasound signal.

In the context of B-ultrasound detection, specific application scenarios, such as the presence of plaques in blood vessels (as shown in Figure 2), necessitate precise control of mechanical pressure and motion amplitude. Excessive pressure can result in the detachment of blood vessel plaques, posing life-threatening risks. Conversely, insufficient amplitude can lead to unclear imaging and poor detection performance. To achieve optimal performance, the mechanical arm must operate within a reasonable range, ensuring that the feedback force remains within safe limits when in contact with the human body. This objective can be accomplished by utilizing an intelligent force feedback control system programmed using a computer or a single-chip microcomputer.

The outcomes of this research hold wide-ranging applications in medical robotics, smart healthcare, remote healthcare, and assisting individuals with disabilities by providing realistic force feedback to the human brain. Additionally, it can assist workers wearing heavy protective clothing in obtaining real-world force feedback.

## 2. Materials and Methods

### 2.1. Materials

Ag nanowire solution (diameter: 45 nm, length: 25 μm, concentration: 1 wt%) was purchased from Kechuang New Material Co., Ltd. (Luoyang, China). Ecoflex, a type of polymer, was purchased from SMOOTH ON, INC. (Macungie, PA, USA). Conductive carbon black (diameter: 30–45 nm) was purchased from XFNANO, INC. (Nanjing, China). PVA (polyvinyl alcohol) was purchased from Shanghai Shifeng Science and Technology Co., Ltd. (Shanghai, China). The nano adhesive used in the experiment was purchased from Shenzhen Nade Adhesive Technology Co., Ltd. (Shenzhen, China). Latex gloves with a thickness of 0.12 mm were purchased from AMMEX Co., Ltd. (Kent, WA, USA).

### 2.2. Methods

Plan A: Ecoflex and conductive nano carbon powder were mixed uniformly at different mass ratios (3:1, 4:1, 5:1, 6:1, 7:1, and 8:1). The mixture was then poured into a stainless steel mold with dimensions of 10 mm × 10 mm × 2 mm. The mold was heated and cured at 60 °C for 1 h. Copper foil electrodes were sealed on both sides of the mold to create a pressure sensor. Various performance indicators were subsequently tested.

Plan B: The optimal mass ratio obtained from Plan A was selected for ecoflex and conductive nano carbon powder. They were mixed uniformly according to this ratio. The resulting mixture was then blended with AgNWs (ethanol dispersion) at different weight ratios (4:1, 6:1, 8:1, 10:1, 12.5:1, 25:1, and 50:1) until fully homogeneous. The mixture was poured into a stainless steel mold with internal dimensions of 10 mm × 10 mm × 2 mm. The mold was heated and cured at 60 °C for 1 h. Copper foil electrodes were sealed on both sides to fabricate the pressure sensor. Various performance indicators were tested.

The resistance change rates of sample pressure sensors from both Plan A and Plan B were measured under a 2 N pressure to determine the optimal formulation.

Plan C: Similar to Plan B, except that AgNWs (ethanol dispersion) was replaced with pure ethanol solution (without AgNWs). All other details remained the same.

Ecoflex is a platinum-catalyzed silicone rubber. It was prepared by mixing A and B gels in a 1:1 ratio and crosslinking them at a specific temperature to obtain a transparent elastic material with defined elasticity. By adding a certain amount of conductive material, resistive pressure sensors could be fabricated. To ensure the performance of the pressure sensor and facilitate resistance measurement, a “sandwich” packaging structure (shown in Figure 3) was designed. It consisted of conductive electrodes on the top and bottom layers, with the elastic material wrapped around the pressure sensor in the middle layer. This structure enhanced the accuracy and precision of pressure sensor testing.

To increase the measurement range, a square through-hole was machined at the center of each sensor, enhancing the sensor’s sensitivity. The hole’s center coincided with the symmetrical center of the elastomeric layer of the sensor. The dimensions of the square through-hole were 1 mm × 1 mm, 2 mm × 2 mm, and 3 mm × 3 mm.

Pressure response was evaluated under different scenarios, and fatigue resistance testing was conducted on selected devices. SEM microscopic structure analysis and EDS spectral analysis were performed on the samples to better understand the relationship between device structure and performance. The pressure sensor was positioned as indicated in Figure 2: at the front end of the handle probe and on the side of the handle to measure the actual pressure load on the probe and the force exerted on the handle side due to friction. During testing, the sensitivity of the antenna’s front end was assessed in various working environments while wearing latex gloves of varying thickness to make the machine’s tactile sensing more akin to human tactile sensing.

## 3. Results

As shown in Figure 4, resistance change rate tests were conducted on samples with different ratios, and the following conclusions can be drawn based on the peak values of the curves. The optimal composition of the pressure sensor in this process was as follows:(ecoflex + carbon powder) to AgNWs (ethanol solution) = 6:1 (mass ratio)ecoflex to carbon powder = 5:1 (mass ratio).

The most sensitive sample in terms of resistance change rate was subjected to 10,000 cycles of fatigue testing, as shown in Figure 5. The sample exhibited good performance. Moreover, it can be observed that after more than 8000 cycles of aging treatment, the performance of the sample is expected to become more stable in future use.

After preparation, the pressure sensors were tested, and the following line graph was obtained. The changes in porous structures were observed in sensors with high and low doping levels.

As shown in Figure 6a–e, the scanning electron microscope images were obtained by scanning samples of ecoflex + carbon powder at a magnification of 10,000 times. Panels a, b, c, d, and e correspond to ecoflex to carbon powder ratios of 3:1, 4:1, 5:1, 6:1, and 7:1, respectively. From the scanning results, it can be observed that carbon black and ecoflex tend to aggregate when mixed, forming a pressure sensor.

As shown in Figure 6f–j, the morphology images obtained at 10,000× and 5000× magnifications are shown for samples with a ratio of (ecoflex + carbon powder): AgNWs ethanol solution = 6:1. Samples f, g, h, i, and j correspond to ratios of (ecoflex + carbon powder): AgNWs ethanol solution of 4:1, 6:1, 8:1, 10:1, and 25:1, respectively. In sample a, slender silver nanowires can be clearly observed, which are uniformly distributed and responsible for the decrease in resistance and increase in stability of the pressure sensor.

As shown in Figure 6k–o, the yellow, blue, green, and red elements represent elemental silver, elemental tin, elemental oxygen, and elemental carbon, respectively. The scanning results of the silver nanowires are consistent with the distribution of the bright thin lines in the image, indicating that these bright thin lines are indeed silver nanowires.

To investigate the role of ethanol in the ecoflex + carbon powder system, five samples with different ratios of ecoflex + carbon powder to ethanol (C_2_H_5_OH) were prepared. As shown in Figure 6p–t, panels p, q, r, s, and t represent ratios of ecoflex + carbon powder to ethanol of 4:1, 5:1, 6:1, 7:1, and 8:1, respectively. The addition of ethanol resulted in varying degrees of pore structure in the samples.

Under the pressure testing condition of 5 kPa, a comparison of the resistance change rate was conducted among samples using the optimal formulation from the three processes. It was evident that the sample of ecoflex-C-AgNWs/ethanol solution exhibited the highest sensitivity. Its sensitivity increased by 19.5% compared to the sample (ecoflex-C) and by 11.3% compared to the sample (ecoflex-C-ethanol). The sample (ecoflex-C-AgNWs/ethanol solution), which only incorporated internal air pore microcracks without through-holes, exhibited sensitive responses to pressures below 5 N. However, with the addition of through-holes, the measurement range of its sensitive response increased to 20 N, representing a 400% increase in the measurement range.

## 4. Discussion

To verify the role of ethanol in the (ecoflex + carbon powder) system, samples with five ratios of (ecoflex + carbon powder): C_2_H_5_OH were prepared. As shown in Figure 6p–t represent (ecoflex + carbon powder): C_2_H_5_OH ratios of 4:1, 5:1, 6:1, 7:1, and 8:1, respectively. After adding ethanol, the samples showed different degrees of porous structures, indicating that the addition of ethanol could cause pores to form in the colloid of (ecoflex + carbon powder), thus improving the sensitivity of the sensor.

Through the analysis of the ratio-dependent changes in resistance of the pressure sensors under a 2 N load for 1 s, we observed that the pressure sensor exhibited optimal performance when the ratio of ecoflex to carbon powder was approximately 5:1. This finding can be attributed to the following reasons.

When the ratio of (ecoflex + carbon powder) is excessively large, the conductivity of the sensor decreases due to the low content of conductive material. This reduction in conductivity hinders the effective transmission of electrical signals, leading to the suboptimal performance of the pressure sensor. On the other hand, when the ratio of (ecoflex + carbon powder) is excessively small, the content of elastic material becomes insufficient, resulting in limited deformation of the pressure sensor under applied pressure. Consequently, the sensitivity and performance of the sensor are compromised.

Therefore, the balanced ratio of ecoflex to carbon powder at approximately 5:1 ensures an optimal combination of conductivity and elasticity, thereby enabling the pressure sensor to achieve the best performance in terms of sensitivity and reliability.

Based on our understanding of the crosslinking mechanism of ecoflex, we found that the crosslinking process involved condensation reactions between silane-functionalized polymers and crosslinkers, resulting in the formation of an elastic material. In order to enhance the sensitivity of the sensor, it was necessary to introduce porous microstructures, which could be achieved through physical or chemical means.

Since crosslinking reactions primarily occur with diols, the addition of an appropriate amount of ethanol to the elastic system can interrupt certain crosslinking reactions and promote the formation of porous structures. However, it should be noted that the addition of ethanol alone would significantly increase the resistance of the sensor. To address this issue, an ethanol solution containing dispersed silver nanowires was introduced. This not only facilitated the creation of porous structures but also ensured stable changes in resistance within the sensor.

By combining the benefits of ethanol-induced porosity and the conductive properties of silver nanowires, we were able to achieve both enhanced sensitivity and consistent resistance changes in the sensor. This approach allowed us to strike a balance between structural modifications and electrical performance, ultimately improving the overall functionality of the pressure sensor.

The microcrack structure and composite conductive mechanism played a significant role in the performance of the pressure sensor. The inclusion of silver nanowires served to enhance conductivity and address issues related to loose and unconsolidated porous structures through the “anchoring effect”. Comparative analysis reveals that sensors incorporating silver nanowires exhibit heightened sensitivity and a wider range of resistance variation, thereby improving the overall capabilities and performance of the sensors, particularly in practical applications.

Based on the observations from the working image in Figure 6, we chose the (ecoflex + carbon powder) ratio of 5:1 as the baseline for our pressure sensor. In order to further investigate the effect of incorporating the ethanol solution of silver nanowires, we explored different component ratios of the pressure sensor by varying the (ecoflex + carbon powder) to AgNW ethanol solution ratio. Specifically, we considered ratios of 4:1, 6:1, 8:1, 10:1, 12.5:1, 25:1, and 50:1, all based on the ecoflex:C ratio of 5:1.

While the pressure sensor demonstrated good performance around the (ecoflex + carbon powder) ratio of 5:1, it exhibited relatively high resistance and small fluctuations during measurement. To assess the impact of incorporating the ethanol solution of silver nanowires, we examined various ratios to identify the optimal configuration for the pressure sensor. This comprehensive investigation allowed us to determine the most suitable (ecoflex + carbon powder) to AgNW ethanol solution ratio, considering both the sensitivity and stability of the sensor.

Based on the analysis of Figure 4, it is evident that the addition of a 1 wt% silver nanowire ethanol solution had a notable positive impact within a specific range. The pressure sensor exhibited its best performance when the ratio of (ecoflex + carbon powder) to AgNWs (ethanol solution) was approximately 6:1. This observation can be attributed to several factors. Firstly, the inclusion of the silver nanowire ethanol solution significantly reduced the resistance of the pressure sensor compared to the configuration without silver nanowires, leading to improved stability of resistance values. Secondly, the presence of ethanol contributed to the formation of a more pronounced pore structure within the sensor. Figure 4 depicts the bottom morphology of (ecoflex + carbon powder) to AgNWs (ethanol solution) at ratios of 4:1 and 50:1, highlighting the variation in pore structures. As the ratio of AgNWs ethanol solution increased, a greater number of pore structures were formed, while lower ratios resulted in fewer pore structures. However, excessive pore structures can compromise the shape integrity of the produced sensor. Therefore, the selection of an appropriate ratio is crucial to ensure optimal performance. For our fatigue testing, we chose the pressure sensor with an (ecoflex + carbon powder) ratio of 6:1, which exhibited a highly stable curve throughout ten thousand fatigue tests, as depicted in Figure 5.

Observations from the samples with different ratios revealed the clustering of ecoflex gel around the AgNWs, which contributed to the reduction of resistance and enhancement of sensor stability. Additionally, the presence of ethanol induced the formation of porous structures within the ecoflex gel, thereby increasing the sensitivity of the pressure sensor. Another factor influencing the decrease in resistance and stability is the uniform distribution of silver nanowires in the colloid.

Upon the addition of ethanol, the samples exhibited varying degrees of pore structure, indicating that ethanol promoted the formation of pores in the ecoflex + carbon powder colloid, consequently improving sensor sensitivity.

When lateral pressure was applied, a certain level of pressure was required to maintain the relative stability of the sensor and probe, as depicted in Figure 7. Consequently, the resistance under positive pressure, added to the baseline resistance value, generally exceeded the resistance under lateral pressure.

In terms of sensitivity, the pressure sensor demonstrated its highest sensitivity at low pressures, specifically within the range of 0–1 N. However, the baseline fixed pressure resulting from lateral pressure exceeded 1 N, leading to lower sensitivity during the initial stage of lateral pressure measurement compared to positive pressure. Nonetheless, in the later stage of applying pressure, the sensitivity of lateral pressure surpassed that of positive pressure.

Figure 7 illustrates that, compared to sensors without AgNWs, sensors with AgNWs exhibited higher sensitivity and a wider range of resistance variation. This improvement effectively expanded the range of the sensor and yielded better performance in practical applications.

## 5. Conclusions

This sample ((ecoflex + carbon powder):AgNWs ethanol solution = 6:1) demonstrated excellent pressure sensing performance and is suitable for applications such as simulating the human hand feel of a handheld handle for B-ultrasound medical robots and researching machine tactile sensing at the probe tip. The introduction of pores enhanced the sensitivity of the sensor, while the incorporation of square-shaped through-holes increased the measurement range, making the flexible sensor samples highly suitable for the machine touch system of B-ultrasound robots. The sensitive and precise measurement results, combined with computer control programs or microcontrollers and image recognition technology, can effectively control the intelligent diagnostic actions of B-ultrasound medical robots, ensuring not only convenience but also safety by accurately controlling the force. Based on the experimental results presented in this paper, these findings can potentially be replicated in other medical robotic applications.

## Figures and Tables

**Figure 1 micromachines-14-01110-f001:**
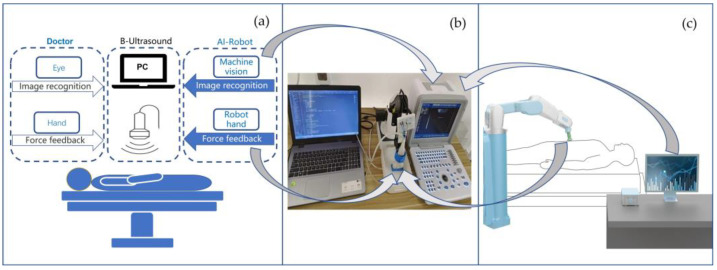
Schematic diagram of the medical robotic system. (**a**) Logical relationship diagram depicting the interplay between machine touch and machine vision, replacing the functions of a doctor’s hands for operation and their eyes for observation. (**b**) System configuration of the experiment, with the flexible sensor enabling machine touch and the computer program facilitating image recognition for machine vision. The devices depicted from left to right are the computer, robotic arm with the sensor, and B-ultrasound equipment. (**c**) Industrial design diagram of an AI-enabled B-ultrasound robot product.

**Figure 2 micromachines-14-01110-f002:**
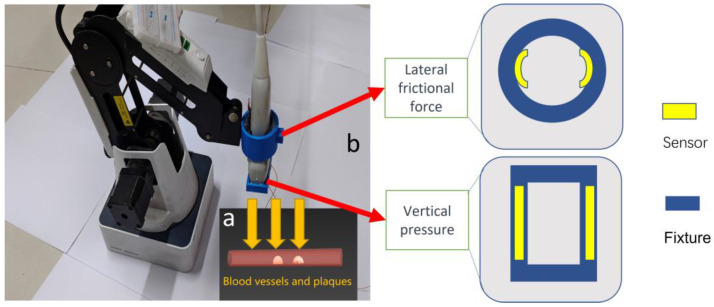
Illustration depicting the force transmission during the detection of plaques and blood clots in blood vessels by the B-ultrasound robot. The image also demonstrates spatial strategies for the arrangement of flexible sensors in different sections. (**a**) Stress state analysis of blood vessels during the descent process of the probe. (**b**) Force testing method design for pressure sensors at different positions on the front and side of the handle. The yellow portion represents the sample of the flexible sensor, and the blue portion represents the fixture that secures the sensor to the probe.

**Figure 3 micromachines-14-01110-f003:**
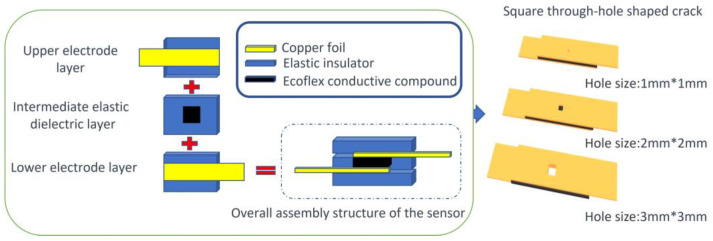
Sensor assembly process and structure, featuring the sandwich packaging structure and the square crack opening structure.

**Figure 4 micromachines-14-01110-f004:**
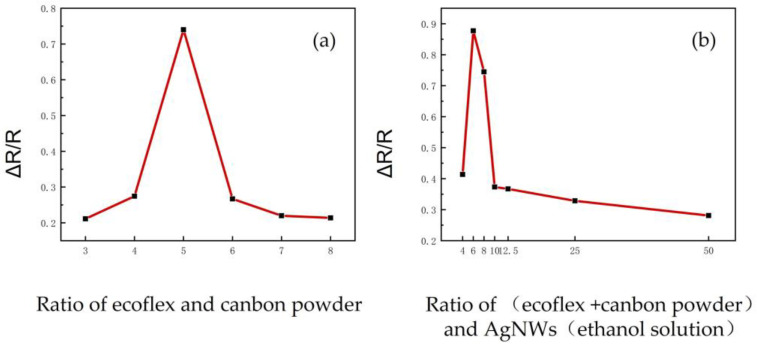
The resistance change rate of the pressure sensor samples for Plan A and Plan B under a pressure of 2 N in 1 s of time.

**Figure 5 micromachines-14-01110-f005:**
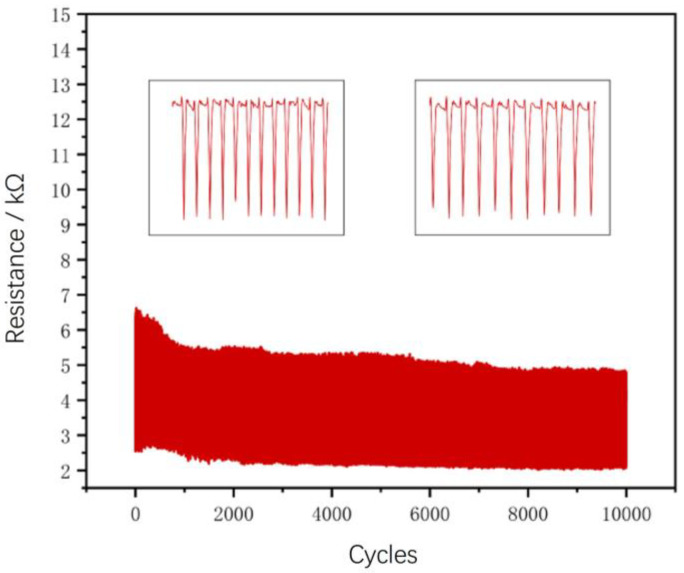
10,000 fatigue tests and local amplification map of the pressure sensor with the optimal formula.

**Figure 6 micromachines-14-01110-f006:**
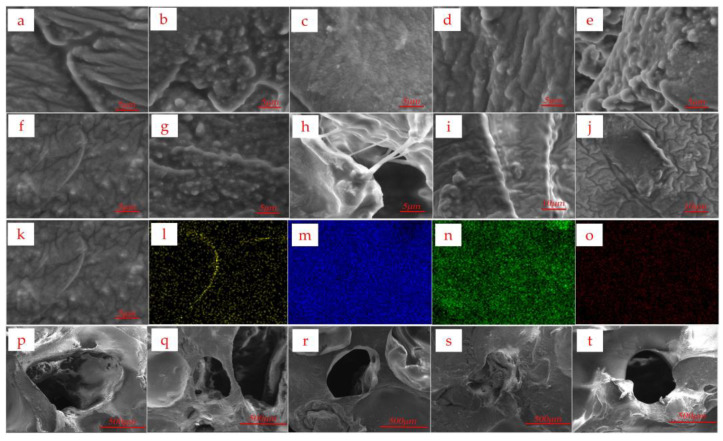
SEM and energy-dispersive X-ray spectroscopy (EDS) elemental distribution map of the samples. (**a**–**e**) SEM images, where (**a**–**e**) correspond to five different sample formulations with ecoflex: carbon powder ratios of 3:1, 4:1, 5:1, 6:1, and 7:1 (mass ratio). (**f**–**j**) SEM images, where (**f**–**j**) correspond to five different sample formulations with (ecoflex + carbon powder): AgNWs (ethanol solution) ratios of 4:1, 6:1, 8:1, 10:1, and 25:1 (mass ratio). (**k**–**o**) SEM and energy-dispersive X-ray spectroscopy (EDS) elemental distribution maps of AgNWs in Plan B samples, where the yellow color indicates the presence of silver element, blue indicates silicon element, green indicates oxygen element, and red indicates carbon element. (**p**–**t**) SEM images, where (**p**–**t**) respectively represent five different formulations of samples with (ecoflex + carbon powder): ethanol ratios of 4:1, 5:1, 6:1, 7:1, and 8:1 (mass ratio).

**Figure 7 micromachines-14-01110-f007:**
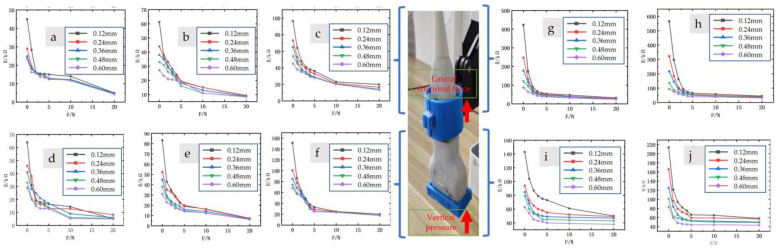
Performance testing of composite porous structure sensors fabricated by ethanol chemical etching and artificial physical pore formation methods. Lateral grip handle force testing with different physical pore size: (**a**) 1 mm × 1 mm; (**b**) 2 mm × 2 mm; and (**c**) 3 mm × 3 mm. Forward force testing of the front end of a probe with different physical pore size: (**d**) 1 mm × 1 mm; (**e**) 2 mm × 2 mm; and (**f**) 3 mm × 3 mm. Encapsulation of pressure sensors through 1–5 layers of latex gloves (thickness range: 0.12, 0.24, 0.36, 0.48, 0.60 mm) to mimic the impact of gloves on the operator’s tactile sensation. Performance testing of composite porous structure sensors fabricated by ethanol chemical etching and artificial physical pore formation methods. The presence of AgNWs in the sample provides mesoscale conductivity mechanisms and enhances the strength of the porous structure with toughening effects. Lateral grip handle force testing with different physical pore size: (**g**) 1 mm × 1 mm; (**h**) 2 mm × 2 mm. Forward force testing of the front end of a probe with different physical pore size: (**i**) 1 mm × 1 mm; (**j**) 2 mm × 2 mm. Encapsulation of pressure sensors through 1–5 layers of latex gloves (thickness range: 0.12 mm, 0.24 mm, 0.36 mm, 0.48 mm, 0.60 mm) to mimic the impact of gloves on the operator’s tactile sensation.

## Data Availability

Not applicable.

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
