# Peer review of "Flexible Pressure Sensors Based on Microcrack Structure and Composite Conductive Mechanism for Medical Robotic Applications"

_micromachines, 2023, doi:10.3390/mi14061110_

Round 1
Reviewer 1 Report
Qiang Zou et al. developed a flexible pressure sensor based on a square through-hole shaped crack that can be used as a machine touch system of the B-ultrasound robot. This application is very important and meaningful. However, if the innovation of this paper is mainly focus on the flexible sensor, the innovation seems not enough. The author carried out many experiments to find the best mix ratio for the sensor, but the innovation is still not clear due to lack of comparison with other existing sensors. Therefore, the author should re-organize this paper to show innovations more clearly.
In addition, some other key questions need to be answered as follows:
1. The authors declare that the flexible pressure sensors are based on microcrack structure. It is improper to describe the sensor by using the word of “microcrack”, because the crack for the sensor in this work is macroscopic with size beyond 1mm as shown in Figure 3. The microcrack usually refer to micro-nano topographic structures of the sensors, e.g. Sun et al. Microsystems & Nanoengineering (2022) 8:111. https://doi.org/10.1038/s41378-022-00419-6.
2. In the introduction part, the authors declare that existing flexible pressure sensors have some limitations. What are the limitations? More details are needed.
3. Compared to existing flexible pressure sensors, what is the advantages of the sensor in this work. A comparison table is needed.
4. Subtitles 2.2 and 3.1.1 should be deleted. Level-3 subheadings seem to be not required in this paper.
5. In the part of Results, it should be re-organized. The current title 3.1 and 3.2 is improper for a research paper. Research paper is not a simple figure stacking and sorting. For this part, the results should be described and summarized and as a scientific logic.
6. In figure 4, the arrows are not needed. The axes have been already there.
7. Figure 6, 7, 8, 9 can be merged in one big figure.
8. The left figures in Figure 2, 10 and 11 are totally same. It is improper.
NA
Author Response
Thank you for your guidance. Please refer to the attached document for the specific details.

Reviewer 2 Report
This work proposed a flexible resistive pressure sensor with optimized composite formula to provide excellent pressure sensing performance for applications such as simulating the human hand feel on B-ultrasound medical robots and the tactile sensing at the probe tip of researching machine. The mechanism design and description are thorough, the experiment was conducted with solid data analysis. In my view, the work can be considered for acceptance with the current form.
Author Response

(The authors gave the same response as above.)

Reviewer 3 Report
This manuscript presents a flexible pressure sensor with a microcrack structure and composite conductive mechanism. Experimental results of the sensor performance are reported. However, this manuscript must be significantly improved based on the following issues:
1.-The English grammar and style must be enhanced.
2.-The abstract should add the main results and conclusions.
3.-The introduction is very weak and short. It did not include the research problem. This section must consider the research problem. In addition, this section must incorporate discussions on the main parameters, advantages, and limitations of other pressure sensors reported in the literature.
4.-The introduction must add the main advantages and scientific contribution of the proposed sensor compared to others reported in the literature.
5.-A Table with the main parameters, advantages, and limitations of the proposed sensor compared to other pressure sensors should be considered.
6.-The section on materials and methods is very short. This section must include a technical description of the working principle of the proposed sensor. Furthermore, schematic views of the operating principle, dimensions, main components, and materials of the proposed sensor should be incorporated. Also, the authors should include more detailed information on the different stages of the fabrication process of the proposed sensor.
7.-The section on results is weak. This section must add more detailed information on the different characteristics of the instruments used in the experimental tests.
8.-Figures 2 and 3 have mistakes in the labels. Always capitalize the first letter of the labels. For instance, lateral frictional force must be changed to Lateral frictional force.
9.-Figure 4 and 5 are not suitable for a scientific article.
10.-Figures 6-9 are very short. The size of these Figures must be increased. In addition, the resolution of these Figures must be improved.
11.-Graphs of figures 10 and 11 have units in the two vertical axes. These figures must have only units in one vertical axis. The size of these Figures must be increased.
12.-The redaction of the discussions is confusing. The authors must significantly enhance the discussions of the main results.
13.-Discussion on the noise should be included.
14.-What is the influence of the temperature and relative humidity of the environment on the performance of the proposed sensor?
15.-The authors should include experimental results of the main performance parameters of the proposed sensor. For instance, the sensitivity and resolution of the proposed sensor.
16.-What is the future research work?
17.-The conclusions must be modified based on the above comments.
The English grammar and style must be enhanced.
Author Response

(The authors gave the same response as above.)

Round 2
Reviewer 1 Report
The authors address most of the issues. It can be accepted in present form.
Reviewer 3 Report
This manuscript was significantly enhanced based on all the comments of the reviewer.
The English grammar is good.